# Genetic origins and proteomic consequences of kinetoplast loss in trypanosomes

Melanie Ridgway[1‡¤], Douglas O. Escrivani[1‡], Markéta Novotná[1], Amy Wood[1], Michele Tinti[1], Achim Schnaufer[2], David Horn[1]*

1 Faculty of Life Sciences, University of Dundee, Dow Street, Dundee, United Kingdom, 2 Institute of Immunology and Infection Research, University of Edinburgh, Edinburgh, United Kingdom,

‡ These authors are Joint first authors.
¤ Current address: Molecular and Biomedical Science, School of Biological Sciences, Adelaide University, Adelaide, South Australia, Australia
* d.horn@dundee.ac.uk

## Abstract

The kinetoplast incorporates the large mitochondrial genome present in the eponymous Kinetoplastida. *Trypanosoma brucei* is an African trypanosome that can lose kinetoplast DNA (kDNA), however, when the nuclear-encoded gamma subunit of the mitochondrial $F_1F_O$-ATP synthase (γATPase) is mutated. These mutations, analogous to a broken camshaft at the core of the ATP synthase rotary motor, are associated with multidrug resistance, and correlated with tsetse-fly independent mechanical transmission, and geographical spread of these parasites beyond Africa. Here we engineer kDNA-independent *T. brucei* to explore origins and consequences of kDNA loss. We use oligo targeting to edit the native *γATPase* gene, and selection with the ATP synthase targeting drug oligomycin to enrich the desired mutants. Using this approach, we identify novel M[282]F, M[282]W, and M[282]Y mutants, and subsequently generate precision-edited strains expressing the previously described L[262]P or A[273]P mutants, or the novel M[282]F mutant. Heterozygous M[282]F mutants retain sensitivity to the kDNA-targeting drug acriflavine, while homozygous M[282]F mutants are acriflavine resistant. Proteomic analysis of the kDNA-positive homozygous M[282]F mutant reveals highly specific depletion of ATP synthase-associated proteins, but not the $F_1$ subunits. Proteomic analysis following acriflavine-induced kDNA loss then reveals depletion of kDNA-binding proteins and mitochondrial RNA-processing factors alongside increased expression of mitochondrial membrane-associated transporters. We conclude that *T. brucei* cells with a homozygous *γATPase* M[282]F mutation remodel ATP synthase subunit expression and readily tolerate kDNA loss, which is accompanied by substantial remodelling of the mitochondrial proteome.

**Data availability statement:** The high-throughput sequencing data generated for this study (amplicon sequencing and whole genome sequencing) have been deposited at the Sequence Read Archive under accession code PRJNA1380964 (https://www.ncbi.nlm.nih.gov/bioproject/1380964). The mass spectrometry proteomics data generated for this study have been deposited at the PRIDE repository under accession code PXD071938 (https://www.ebi.ac.uk/pride/archive/projects/PXD071938).

**Funding:** This work was supported by a Wellcome Centre Award (223608/Z/21/Z), DH was co-applicant, and a Wellcome Investigator Award to D.H. (217105/Z/19/Z). The funder played no role in the study design, data collection and analysis, decision to publish, or preparation of the manuscript.

**Competing interests:** The authors have declared that no competing interests exist.

## Author summary

Mutations in the gamma subunit of the mitochondrial ATP synthase in parasitic African trypanosomes can have major consequences. Specifically, the entire large and complex mitochondrial genome, the kinetoplast DNA (kDNA), is rendered dispensable, and the cells become resistant to important kDNA targeting drugs. Veterinary parasites with these mutations have also spread outside Africa through simple mechanical transmission, either sexually or by biting flies or vampire bats. Here, we precision-edit the gamma subunit to replicate previously described mutants and identify a novel mutant that readily tolerates kDNA loss. Using quantitative proteomics, we demonstrate highly specific depletion of ATP synthase-associated proteins pre kDNA loss. We then use genome sequencing to show that the kDNA can be completely lost by these cells and demonstrate that cells lacking mitochondrial nucleic acids display specific depletion of mitochondrial nucleic acid-binding proteins. Notably, several mitochondrial membrane-associated transporter complexes are increased in abundance. Thus, we establish a method to test precise γATPase mutations and to identify new mutations associated with kDNA loss. We also show that trypanosomes with dispensable kDNA specifically remodel expression of ATP synthase subunits pre kDNA loss and substantially remodel the mitochondrial proteome post kDNA loss.

## Introduction

*Trypanosoma brucei brucei* is an African trypanosome that is transmitted by tsetse flies, causing nagana disease in cattle and other livestock. Closely related and similarly transmitted African trypanosomes cause sleeping sickness in humans. These parasites are kinetoplastids, flagellated protozoa that contain their mitochondrial genome (mtDNA) in a kinetoplast, hence called kinetoplast DNA, or kDNA. The kDNA is a cytologically prominent feature and comprises a huge network of approximately twenty-five maxicircles and thousands of minicircles, encoding eighteen protein subunits of the mitochondrial respiratory chain, the $F_1F_O$-ATP synthase and the mitoribosome, as well as ribosomal RNA and RNA-editing associated guide RNAs [1,2]. Insect stage *T. brucei* depend on kDNA-encoded proteins for oxidative phosphorylation, while the bloodstream stage requires $F_1F_O$-ATPase activity to generate the mitochondrial membrane potential; whereby ATP hydrolysis by $F_1$ is coupled to proton transfer by $F_O$ [3]. Key to this coupling is the central $F_1$ γ subunit (γATPase), which acts like a camshaft. γATPase is mechanically coupled to the membrane embedded ring composed of 10 *c* subunits that, together with the A6 subunit (also known as the *a* subunit), forms the proton translocating part of $F_O$ [4,5]. A6 is the only $F_1F_O$-ATPase subunit encoded in kDNA. Because of its essentiality to African trypanosomes and its unique properties, kDNA has proven to be an excellent drug target, albeit with challenges associated with resistance [6,7].

Remarkably, African trypanosomes of the *T. brucei* group can lose their kDNA, and *T. b. equiperdum and T. b. evansi*, first identified in the 1800's, present two examples [8,9]. These parasites still infect equids and various mammals, respectively, and grow as bloodstream forms [7] but cannot differentiate to tsetse insect stages [10]. Although unable to complete the usual life cycle in tsetse, they are transmitted mechanically, either sexually in equids (*equiperdum*), or by biting flies or vampire bats (*evansi*). Consequently, *T. b. equiperdum* and *T. b. evansi* cause diseases known as dourine and surra, respectively, that have spread beyond tsetse endemic regions in Africa, extending to Asia, South America and parts of Europe [11].

kDNA dispensability is caused by specific mutations in γATPase, by allowing for generation of a mitochondrial membrane potential, albeit potentially reduced, in the absence of the kDNA-encoded $F_O$ subunit A6 [12–14]. This is thought to involve mechanical uncoupling of the $F_1$ and $F_O$ components, enhanced ATP hydrolysis by $F_1$ and electrogenic exchange of mitochondrial $ADP^{3-}$ for cytosolic $ATP^{4-}$ by the mitochondrial ADP/ATP carrier [15]. Remarkably, beyond the A6 subunit, and the mitoribosome subunits required for its translation, no other kDNA-encoded protein appears to be specifically required to maintain the viability of wild-type bloodstream-form *T. b. brucei*. Trypanosomes with γATPase mutations have emerged several times independently, being equivalent to mutations that enable mtDNA loss in petite-negative yeast [8,14,15]. These parasites are either dyskinetoplastic or akinetoplastic, lacking some or all of their kDNA, respectively. Other subsequent changes appear to have facilitated adaptation to a tsetse fly independent life-cycle [9].

The kDNA has proven to be an excellent drug target, and several veterinary anti-trypanosomal drugs target kDNA, including ethidium bromide and isometamidium. kDNA loss or dispensability in *T. b. evansi, T. b. equiperdum*, and other γATPase mutants renders these cells multidrug resistant, however [7]. Despite connections to the parasite life cycle, geographical disease distribution, and drug resistance, the mechanisms linking γATPase mutations to kDNA dispensability are not fully understood. To develop our understanding of the origins and consequences of kDNA loss in trypanosomes, we used oligo-targeting [16] and engineered kDNA-independent kinetoplastids. We introduced novel and known mutations in the native *γATPase* gene and found that a novel homozygous $M^{282}F$ edit rendered the kDNA dispensable. We then used proteomics analysis to assess the complement of proteins impacted by γATPase mutation pre and post kDNA loss, revealing specific impacts on the mitochondrial ATP synthase, mitochondrial nucleic acid binding proteins and mitochondrial membrane-associated transporters.

## Results

### γATPase editing yielded known and novel oligomycin-resistant mutants

Three distinct non-synonymous substitutions have been identified in the γATPase subunit in trypanosomes that display kDNA dispensability; $L^{262}P$, $A^{273}P$, and $M^{282}L$ [12–14]. $A^{273}P$ was identified as a homozygous single-nucleotide mutation in *T. b. equiperdum*, $M^{282}L$ was identified as a heterozygous single-nucleotide mutation in some isolates of *T. b. evansi* [14], and $L^{262}P$ was later identified as a homozygous single-nucleotide mutation in *T. b. brucei* following acriflavine-selection in the laboratory [12]. The link to kDNA dispensability was validated for both the $L^{262}P$ and $A^{273}P$ mutations using ectopic *γATPase* expression in transgenic *T. b. brucei*, but a similar assay failed to validate the $M^{282}L$ mutation [12].

To assess the impact of specific mutations at the native *γATPase* gene locus (Tb927.10.180) in *T. b. brucei*, we used oligo targeting for precision editing [16], followed by oligomycin selection to enrich those mutants that become independent of the $F_O$ component of the ATPase (Fig 1A); oligomycin targets the proton-binding $F_O$ subunit *c* [17]. Given uncertainty regarding the impact of the $M^{282}L$ mutation, we began by assessing edits at this site. For oligo targeting, we typically deliver approximately 50 base 'reverse-strand' single-stranded oligodeoxynucleotides (ssODNs) by electroporation, and in this case, we designed a 53-b ssODN to target the $M^{282}$ site with a centrally located and degenerate 'NNN' (N=A, C, T, G) codon (Sheet 1 in S1 Data). Wild type *T. b. brucei* cells were transfected in duplicate and grown with oligomycin at 200 nM; approximately three times the $EC_{50}$ (Effective Concentration of drug to inhibit growth by 50%). We then extracted genomic DNA from surviving cells after six days, PCR-amplified the edited region in the *γATPase* gene, deep-sequenced

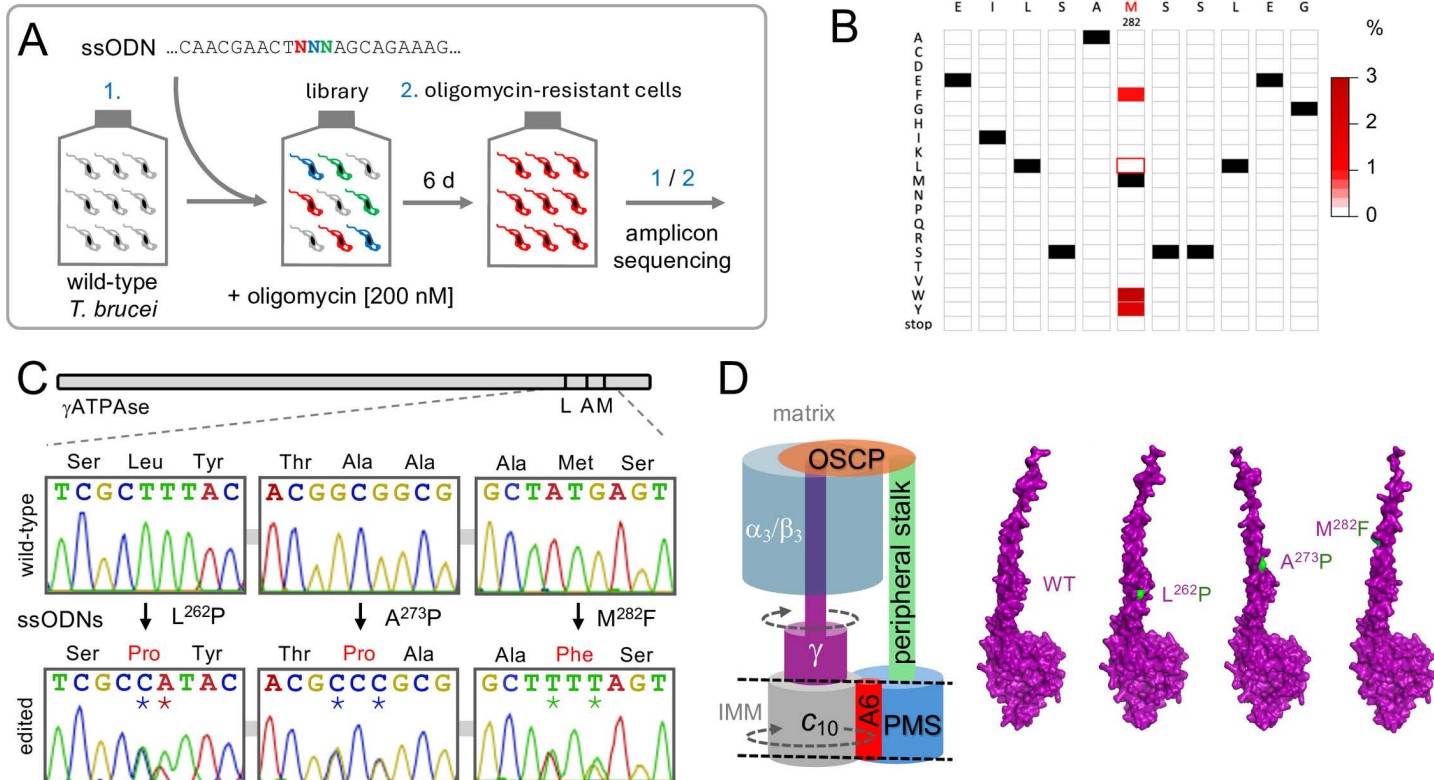

**Fig 1. γATPase editing yielded known and novel oligomycin-resistant mutants. (A)** The schematic illustrates oligo targeting for saturation mutagenesis of the *T. b. brucei* γATPase M[282] residue. A sixty-four fold degenerate ssODN was transfected into wild-type *T. b. brucei* cells, followed by oligomycin selection and γATPase amplicon-sequencing. **(B)** The heat map shows relative representation of each possible amino acid variant at the targeted M[282] site and at adjacent sites; averages for two independent oligomycin-resistant cultures relative to an unedited control. More than 8 M reads were mapped per site on average. Unedited codons are indicated (black) as is the previously reported M[282]L mutation (red outline). **(C)** The Sanger sequencing traces show single allele edits encoding the L[262]P, A[273]P and M[282]F mutations, each involving a double nucleotide edit; edited nucleotides are indicated by asterisks. **(D)** Simplified schematic of the trypanosome $F_1F_O$-ATP synthase and key components discussed here. Names of $F_1$ subunits are in white letters ($F_1$ subunits p18, delta and epsilon were omitted for simplicity). The α/β hexamer (cyan) is held in place by attachment to the peripheral 'stator' stalk (green) via the OSCP subunit (orange). The proton-translocating part consists of the $c_{10}$-ring (grey) and the kDNA-encoded A6 subunit (red). PMS, peripheral membrane subcomplex (blue); IMM, inner mitochondrial membrane. The AlphaFold models for wild-type (WT) and mutant γATPase were generated using the AlphaFold server, showing mutant residues in green.

the *γATPase* amplicons (Fig 1A), and quantified variant codons. The heatmap in Fig 1B shows relative representation of alternative codons at the targeted site and at flanking sites following oligomycin selection. The analysis revealed highly specific editing at the targeted site, and multiple M[282] *γATPase* edits enriched in the oligomycin-resistant population, all of which encode aromatic residues, M[282]F, M[282]W, and M[282]Y (Fig 1B). Notably, all of the enriched mutants required double or triple nucleotide edits while the naturally occurring M[282]L mutation of uncertain significance, accessible via two distinct single nucleotide edits, or four distinct double nucleotide edits, was not enriched.

We next designed specific ssODNs to introduce one of the novel edits identified above, M[282]F[TTT], or the other non-synonymous substitutions previously linked to kDNA dispensability, L[262]P[CCA] and A[273]P[CCC] (Fig 1C, Sheet 1 in S1 Data); a double nucleotide edit in each case allowed us to distinguish between true edits and spontaneous mutations, the vast majority of which are limited to single nucleotides. We transfected wild type *T. b. brucei* cells with each ssODN, selected the cells with oligomycin at 200 nM, and sub-cloned the resistant cells that emerged. We extracted genomic DNA from the sub-clones, PCR-amplified the edited region in the *γATPase* gene, and Sanger sequenced the amplicons. Sequencing

analysis revealed that all three heterozygous edits were effectively introduced ([Fig 1C]). We then used AlphaFold [18] to visualise how these edits may impact γ-subunit function. The $F_1$ component of the ATP synthase comprises a rotor made up of three α-subunits and three β-subunits with a central γ-subunit, which is analogous to a camshaft ([Fig 1D]). The models predict conformational defects associated with each γATPase mutation, in the extended α-helical camshaft-like segment, and more clearly apparent in the $A^{273}P$ mutant. These mutations may interfere with interaction with the α/β hexamer, perhaps uncoupling $F_1$ from $F_O$ as described for mitochondrial genome integrity mutations in yeast [19], thereby also reducing sensitivity to the $F_O$ inhibitor oligomycin. Thus, *γATPase* precision editing yielded known and novel heterozygous oligomycin-resistant mutants.

### Only bi-allelic $M^{282}F$ *γATPase* editing yielded acriflavine-resistant mutants

Prior analyses suggested that γATPase mutant dosage may be important. Specifically, a heterozygous $A^{281}Δ$ mutant γATPase allele was reported to be preferentially expressed in *T. b. evansi* [14], while both $L^{262}P$ and $A^{273}P$ γATPase substitutions with a validated link to kDNA dispensability are present as homozygous mutations in *T. b. brucei* [12] and *T. b. equiperdum* [14], respectively. Since differential expression of mutant alleles could impact the behaviour of heterozygous mutants, we favoured the analysis of homozygous mutants. Although we had not previously observed homozygous editing using oligo targeting, we identified a homozygous *γATPase* $M^{282}F^{TTT}$ edited clone following oligomycin selection as detailed above. Indeed, a synonymous polymorphism present seven codons downstream of the targeted codon allowed us to show that both heterozygous and homozygous $M^{282}F^{TTT}$ strains remained diploid at this locus, having retained both *γATPase* alleles ([Fig 2A]). Since oligomycin, used above to enrich for edited cells, targets the $F_O$ subunit of the ATPase

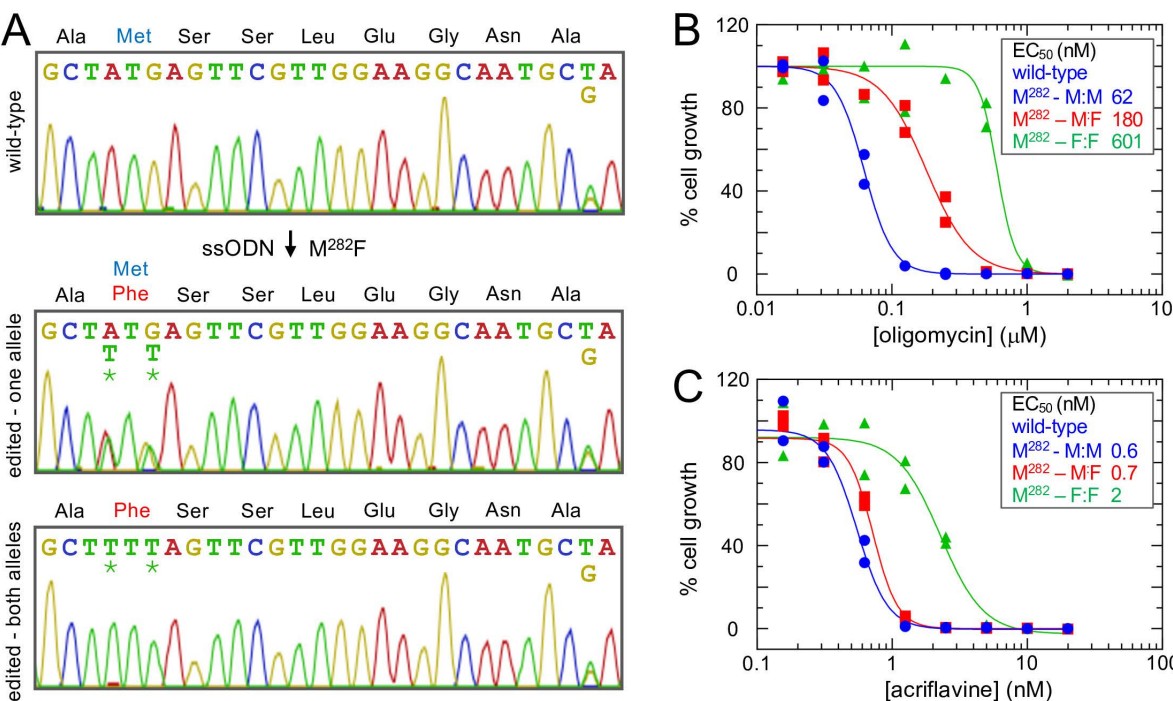

**Fig 2. Only bi-allelic *γATPase* $M^{282}$ editing yielded acriflavine-resistant mutants. (A)** The Sanger sequencing traces show *T. b. brucei* γATPase $M^{282}F$ edits, both heterozygous and homozygous. A GCT/G, alanine polymorphism can be seen on the right-hand side of each panel, confirming retention of both alleles. Edited nucleotides are indicated by asterisks. **(B)** Dose-response curves for oligomycin, measured in duplicate. $EC_{50}$ values are shown. **(C)** Dose-response curves for acriflavine, measured in duplicate. $EC_{50}$ values are shown.

rather than kDNA, we determined whether the edited mutants retained kDNA. Indeed, more than 99% of heterozygous L$^{262}$P and A$^{273}$P mutant cells and homozygous M$^{282}$F mutant cells remained kDNA positive as assessed by DNA-staining and microscopy.

Access to both heterozygous and homozygous M$^{282}$F$^{TTT}$ strains presented an opportunity to compare impacts on oligomycin sensitivity. We performed dose response assays comparing wild-type, heterozygous and homozygous mutants. An oligomycin dose response assay revealed that heterozygous M$^{282}$F$^{TTT}$ parasites displayed 3-fold increased EC$_{50}$ while homozygous M$^{282}$F$^{TTT}$ mutant parasites displayed 10-fold increased EC$_{50}$ (Fig 2B); the heterozygous L$^{262}$P and A$^{273}$P mutants displayed 8-fold and 22-fold increased EC$_{50}$, respectively (S1 Fig). Thus, all of the $\gamma ATPase$ edits yielded significant increases in oligomycin resistance ($P < 1e^{-4}$), as expected, but the dosage of mutant $\gamma ATPase$ alleles in the M$^{282}$F$^{TTT}$ edited cells impacted the relative shift in EC$_{50}$.

We next performed dose response assays using acriflavine, a DNA-intercalating agent that targets kDNA, again comparing wild-type, heterozygous and homozygous mutants (Fig 2C). We observed that homozygous M$^{282}$F$^{TTT}$ parasites displayed a significant, 3-fold increased EC$_{50}$ ($P < 1e^{-4}$), while heterozygous M$^{282}$F$^{TTT}$ parasites displayed only 1.1-fold ($P = 0.4$) increase in EC$_{50}$ (Fig 2C); the heterozygous L$^{262}$P and A$^{273}$P mutants both displayed 2-fold and 2.2-fold increased EC$_{50}$ respectively (S1 Fig). Thus, homozygous M$^{282}$F$^{TTT}$ edits yielded acriflavine resistant cells, while heterozygous edits failed to do so, indicating that this mutation is recessive with respect to acriflavine-resistance. We concluded that both heterozygous and homozygous $\gamma ATPase$ M$^{282}$F$^{TTT}$ editing conferred oligomycin-resistance, albeit to differing degrees, while only homozygous $\gamma ATPase$ M$^{282}$F$^{TTT}$ editing conferred acriflavine-resistance.

## ATP synthase remodelling and kDNA loss in homozygous $\gamma ATPase$ mutants

To further elucidate the mechanism underpinning oligomycin and acriflavine cross-resistance, wild-type and homozygous $\gamma ATPase$ M$^{282}$F$^{TTT}$ edited parasites were assessed using high resolution quantitative proteomics on an Orbitrap Astral mass spectrometer with data-independent acquisition (Sheet 2 in S1 Data). The analysis revealed highly specific depletion of all eighteen known nuclear-encoded subunits of the F$_O$ component of the *T. b. brucei* ATP synthase (Fig 3, see Fig 1D); the peripheral stalk proteins, including the oligomycin sensitivity conferring protein OSCP, membrane region proteins and peripheral membrane subcomplex proteins [5]. In striking contrast, subunits of the F$_1$ component of the ATP synthase, including the mutated γ subunit itself, were not depleted (Fig 3A). Thus, proteomic analysis revealed highly specific depletion of subunits of the F$_O$ component of the *T. b. brucei* ATP synthase in homozygous M$^{282}$F$^{TTT}$ mutants. To determine whether depletion of F$_O$ subunits had major impacts on kDNA or on mitochondrial membrane potential, we examined both wild-type cells and homozygous M$^{282}$F$^{TTT}$ mutants by microscopy following DNA-staining, and by flow cytometry following MitoTracker staining. More than 99% of these cells were kDNA positive by microscopy (Fig 3B), and MitoTracker staining appeared unperturbed when assessed using flow cytometry (Fig 3C).

Since acriflavine inhibits kDNA replication and segregation, acriflavine resistance displayed by homozygous $\gamma ATPase$ M$^{282}$F$^{TTT}$ edited parasites suggested that kDNA would be dispensable in these cells. To induce kDNA loss, we grew two parallel cultures of homozygous M$^{282}$F$^{TTT}$ edited cells in the presence of a sub-EC$_{50}$ dose of acriflavine for 7 days (1.25 nM; the EC$_{50}$ is 2 nM, see Fig 2C), which yielded populations containing approximately 50% kDNA negative cells, as determined by DNA-staining and microscopy. Each population was then cloned by limiting dilution in the absence of acriflavine and, when sufficient cells were available after 7–8 days, clones were assessed by DNA-staining and microscopy. Clones that appeared to lack kDNA, two from each independent culture, were selected for further analysis (see Fig 4A). Notably, kDNA negative M$^{282}$F$^{TTT}$ cells displayed a growth defect relative to the kDNA positive parent, with doubling time increased by approximately 1.5-fold to 9.4 h +/-1.5 h (n = 4); parent doubling time was 6.4 h.

We next considered a more sensitive approach to determine whether kDNA had been completely lost, and subjected wild-type cells, kDNA positive M$^{282}$F$^{TTT}$ edited cells, and all four kDNA negative clones, to whole genome sequencing. We used a recent *T. b. brucei* maxicircle and minicircle DNA assembly for the same strain used in our study [2] as a template

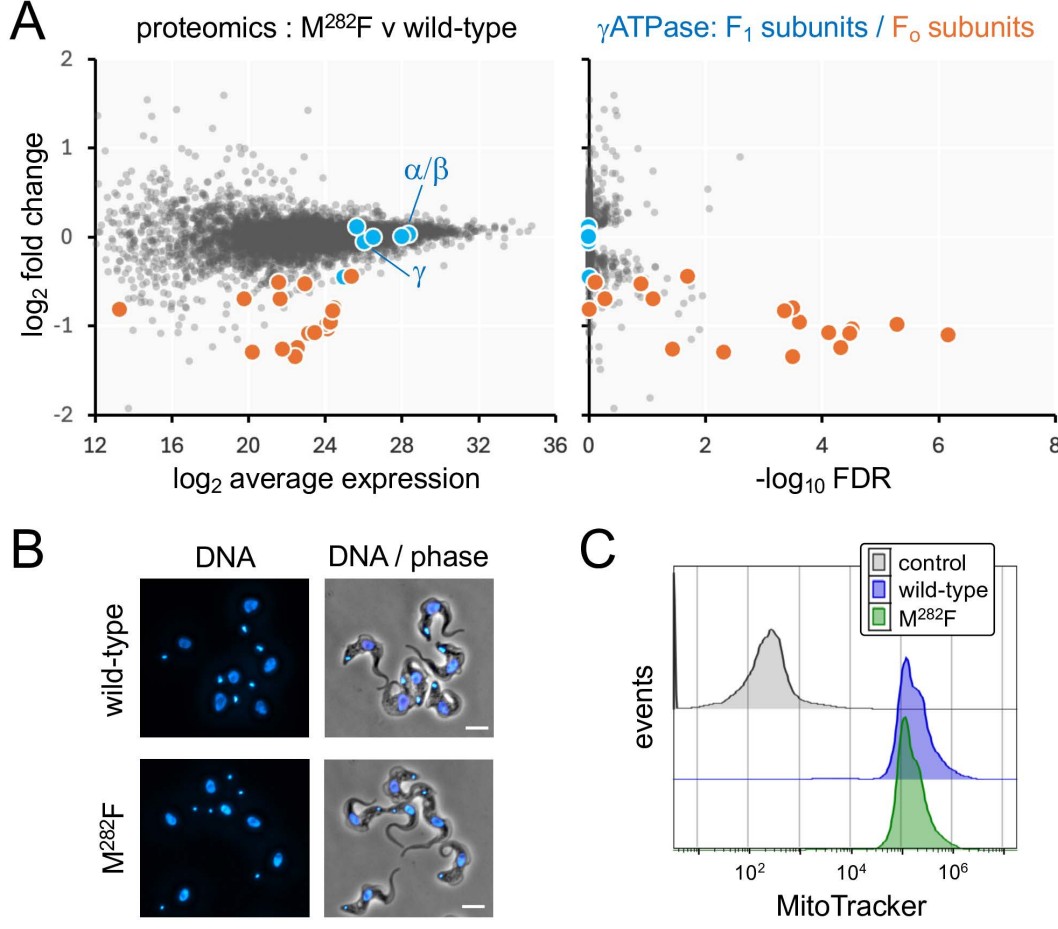

**Fig 3. ATP synthase remodelling in homozygous γATPase mutants. (A)** Proteomics analysis of wild-type *T. b. brucei* and homozygous γATPase M$^{282}$F mutants. Subunits of the F$_1$ and F$_o$ γATPase components are highlighted. Averages from three replicates; n = 6847 proteins. **(B)** The microscopy images show nuclei (larger structures) and kDNA (smaller structures) in wild-type *T. b. brucei* and homozygous γATPase M$^{282}$F mutant cells; DNA was stained with DAPI. Scale bars, 5 μm. **(C)** Flow cytometry analysis of MitoTracker-stained cells, wild-type and homozygous γATPase M$^{282}$F mutants. The data are representative of three technical replicates. Control, unstained cells.

for this analysis and observed highly specific loss of both classes of kDNA in independently generated kDNA negative clones without apparent changes in the nuclear genome (Fig 4B). Closer inspection of minicircle abundance indicated that some were already depleted in M$^{282}$F$^{TTT}$ edited cells prior to acriflavine exposure, with some of the lower abundance mini-circles apparently lost entirely (Fig 4C) Thus, genome sequencing revealed some minicircle loss in homozygous M$^{282}$F$^{TTT}$ mutants pre acriflavine-exposure, and complete elimination of kDNA induced by acriflavine in these cells.

## Mitochondrial proteome remodelling following kDNA loss

To explore the consequences of kDNA loss, we again used high resolution quantitative proteomics to compare homozygous γATPase M$^{282}$F$^{TTT}$ mutants with or without kDNA. Analysis of these proteomes (Sheets 3–6 in S1 Data) revealed substantial changes, including further specific depletion of subunits of the F$_O$ component of the *T. b. brucei* ATP synthase, except for subunit *c*, which displayed increased abundance. As above (Fig 3A), subunits of the F$_1$ component of the ATP synthase, including the mutated γ subunit itself, were not depleted (Fig 5A, S2 Fig). DNA and MitoTracker staining

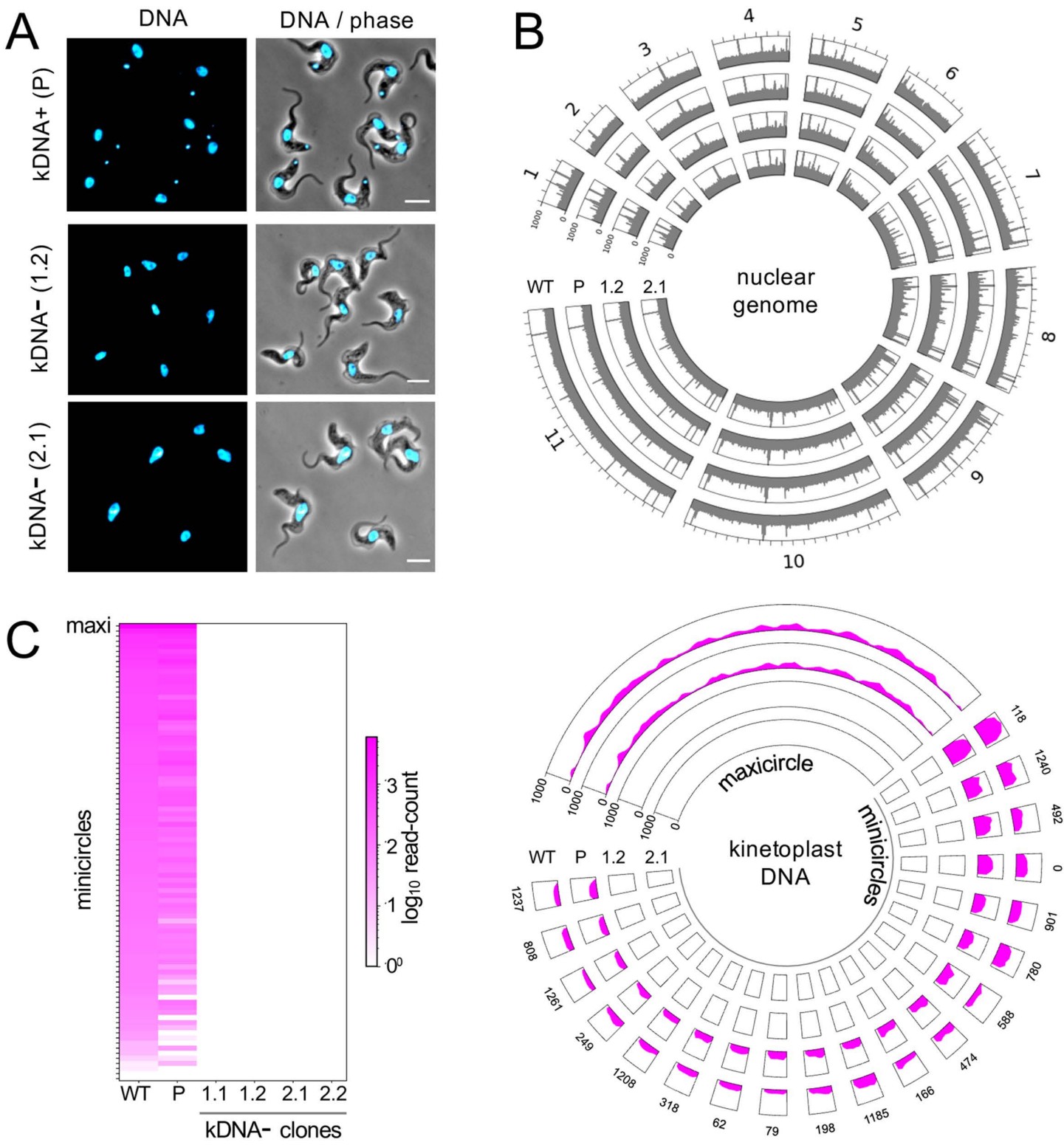

**Fig 4. kDNA loss in homozygous γATPase mutants. (A)** The microscopy images show homozygous γATPase M²⁸²F mutant cells with or without kDNA, the smaller blue DNA-stained structures. DNA was stained with DAPI. Scale bars, 5 µm. **(B-C)** Whole genome sequencing data for wild-type (WT) *T. b. brucei*, the homozygous γATPase M²⁸²F mutant with kDNA (P for parent) and independently generated clones lacking kDNA. **(B)** The upper circular

plot shows genome sequencing data mapped to the *T. b. brucei* nuclear chromosomes 1–11 (grey). The lower circular plot shows genome sequencing data mapped to the *T. b. brucei* kDNA (magenta), maxicircle sequence and the most abundant minicircle sequences; the numbers indicate minicircle ID. Mapping is for 150-bp bins and for two independently generated kDNA negative clones. **(C)** The heatmap shows data for maxicircle sequence, additional minicircle sequences (n=90), and for all four kDNA negative clones. kDNA negative clone numbers are indicated in each panel.

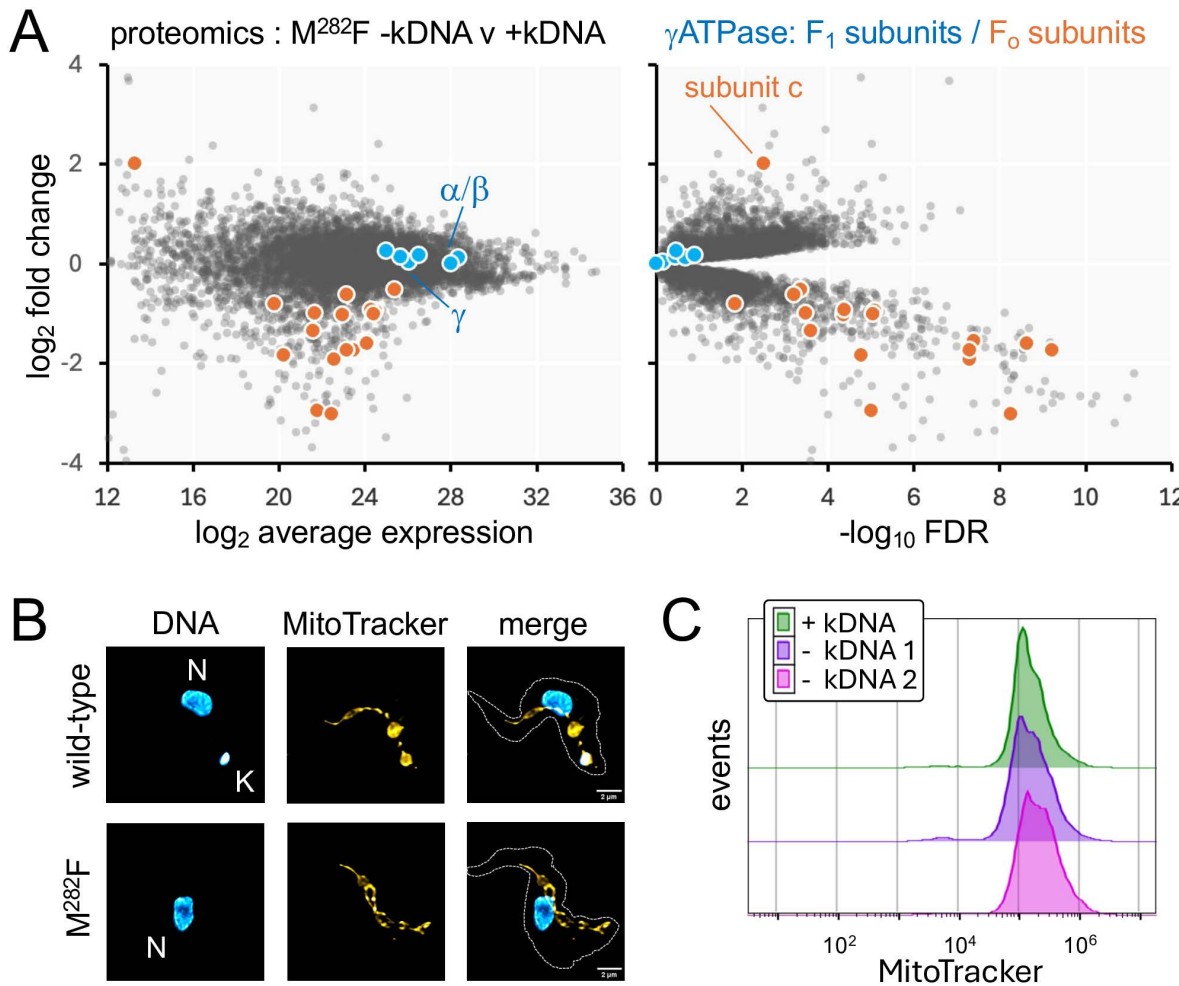

**Fig 5. Proteomic and mitochondrial impacts of kDNA loss. (A)** Proteomics analysis of homozygous γATPase $M^{282}F$ mutants with and without kDNA with subunits of the $F_1$ and $F_o$ γATPase components highlighted. Averages from three replicates; n=6768 proteins; clone 1.2 is shown here and three other clones are shown in Supplementary Fig 2. **(B)** The representative super resolution microscopy images show wild-type *T. b. brucei* and homozygous γATPase $M^{282}F$ mutant cells lacking kDNA, following growth in acriflavine. DNA was stained with DAPI (cyan) and mitochondria were stained with MitoTracker (yellow). Nuclear DNA (N) and kDNA (K) are indicated. Scale bars, 2 µm. A gallery of additional images is shown in Supplementary Fig 3. **(C)** Flow cytometry analysis of MitoTracker-stained cells, homozygous γATPase $M^{282}F$ mutants with or without kDNA. Data are shown for two independent biological replicates without kDNA and are representative of three technical replicates in each case.

followed by super resolution microscopy confirmed complete loss of kDNA in the $M^{282}F^{TTT}$ mutants following acriflavine treatment and revealed broadly maintained mitochondrial structure (Fig 5B, S3 Fig). We also assessed these MitoTracker stained cells by flow cytometry, which suggested that mitochondrial membrane potential was maintained following kDNA loss (Fig 5C).

We next examined changes in the abundance of nuclear and mitochondrial proteins following kDNA loss and found that mitochondrial proteins were selectively and significantly reduced in abundance in all four kDNA negative clones (Fig 6A). More proteins reported a highly significant (-$\log_{10}$ False Discovery Rate [FDR] >4) reduction in abundance relative to proteins that reported increased abundance following kDNA loss (168 v 51), and a closer inspection of >2-fold depleted proteins revealed kDNA-binding proteins and mitochondrial RNA-processing factors (Fig 6B), consistent with destabilisation of these proteins after loss of all mitochondrial DNA and RNA. These included the kDNA-associated proteins involved in DNA compaction [20], and kDNA anchoring to the tripartite attachment complex [21], primases [22], polymerases [23], topoisomerase involved in DNA replication, and mRNA polyadenylation factors [24]. Other depleted proteins were the calcium uniporter, known to interact with subunit *c* of the ATP-synthase [25], and PUF9, an RNA-binding protein involved in

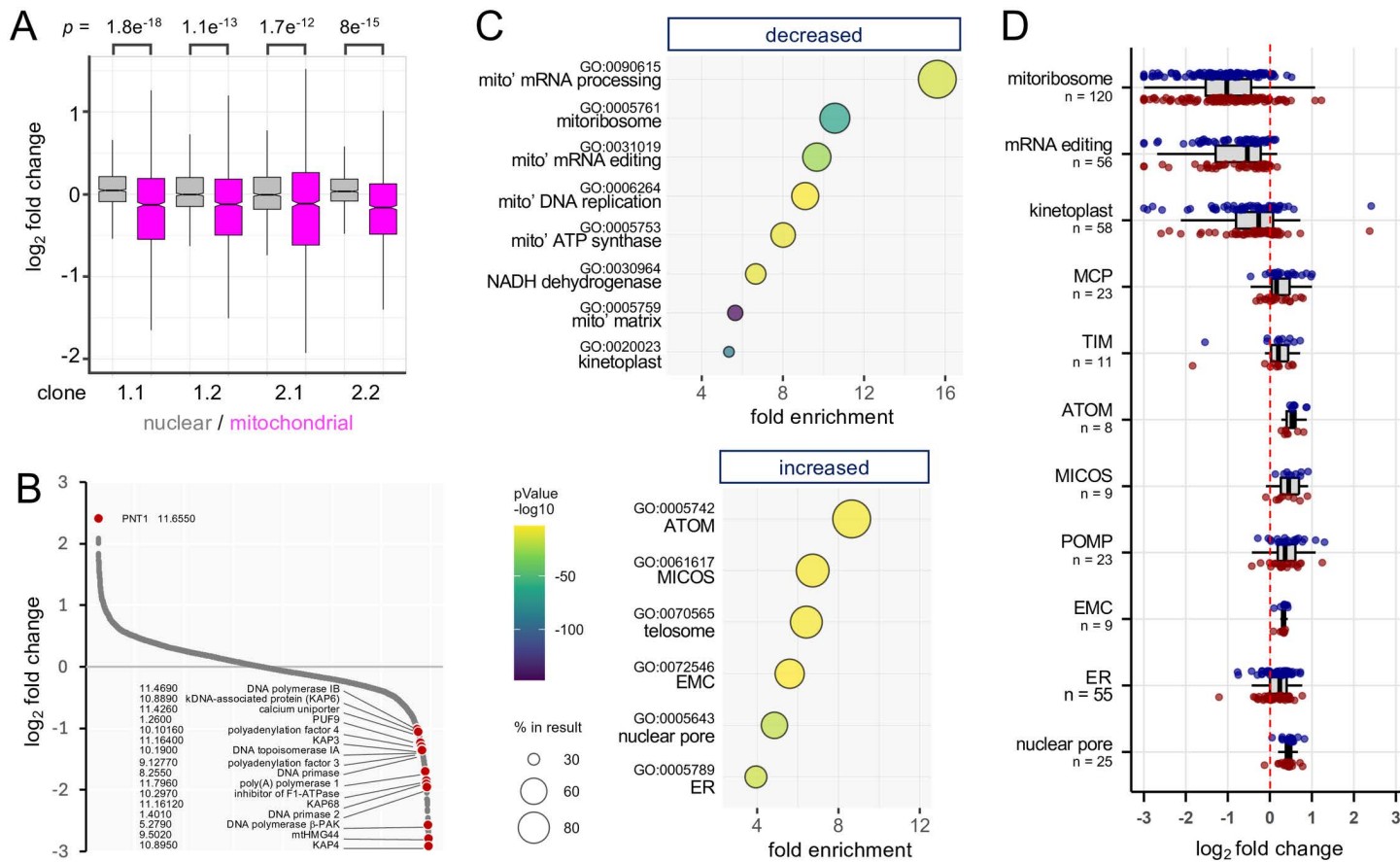

**Fig 6. Mitochondrial proteome remodelling following kDNA loss. (A)** Proteomics analysis of homozygous γATPase M²⁸²F *T. b. brucei* mutants and all four M²⁸²F clones lacking kDNA. The boxplot shows nuclear proteins (n = 1185, GO:0005634) and mitochondrial proteins (n = 1431, GO:0005739). Boxes indicate the interquartile range (IQR) and the whiskers show the range of values within 1.5 × IQR. **(B)** Proteomics analysis showing all proteins with $\log_2$ average expression >16 and with some notable proteins highlighted. **(C)** Gene Ontology profiles for proteins with $\log_2$ average expression >16 that are significantly (>2 –$\log_{10}$ FDR) decreased or increased in abundance in kDNA negative cells. Mito', mitochondrial. **(D)** Selected cohorts of proteins that are significantly decreased or increased in abundance in kDNA negative cells. Boxes indicate the interquartile range (IQR) and the whiskers show the range of values within 1.5 × IQR. MCP, mitochondrial carrier proteins; TIM, Translocases of the Inner Membrane; ATOM, Archaic Translocase of the Outer Membrane; MICOS, Mitochondrial contact site and Cristae Organizing System; POMP, Present in the Outer mitochondrial Membrane Proteome; EMC, ER-Membrane Complex; ER, Endoplasmic Reticulum. Cohorts were derived using GO-terms except for kinetoplast-associated proteins from [31] and MCP, TIM and POMP, derived using wild-card searches, MCP*, TIM* and POMP* at https://tritrypdb.org [43]. Data are shown for kDNA negative clones 1.2 (blue) and 2.1 (red).

nuclear cell cycle regulation [26]. Notably, IF1, an inhibitor of $F_1$-mediated ATP hydrolysis, was also depleted. Expression of this protein was thought to be restricted to the insect stage, where ATP synthesis, but not hydrolysis, is essential [27]. This observation might indicate increased ATP hydrolysis following kDNA loss. In contrast, PUF9 target 1 (PNT1), a kDNA replication-associated peptidase [28], is notably increased in abundance in kDNA negative cells (Fig 6B).

Finally, we profiled proteins that were significantly (FDR < 0.01) reduced (n = 507) or increased (n = 761) in abundance following kDNA loss (Sheets 3–6 in S1 Data), using Gene Ontology (GO) terms. The top GO-term hits for depleted proteins were exclusively associated with the mitochondrion, again including DNA and RNA binding proteins, and ATP synthase (Fig 6C, upper panel). Kinetoplast ($P = 1.4e^{-89}$) and mitochondrial matrix ($P = 4.1e^{-148}$) registered the highest significance, and components of the NADH dehydrogenase, complex I of the electron transport chain, were also significantly depleted; perhaps unsurprisingly since several components of this complex are encoded in kDNA [1,2]. In contrast, mitochondrial membrane-associated transporters, and endoplasmic reticulum (ER) associated proteins, were significantly increased in abundance (Fig 6C, lower panel). The telomere-telomerase complex and nuclear pore proteins were also increased, perhaps reflecting connections between nuclear and kDNA replication [29]. Changes in abundance for several of these cohorts of proteins are shown in Fig 6D, revealing substantial depletion of the mitochondrial ribosome, RNA editing complex [30] and kinetoplast-associated proteins [31]. The tripartite attachment complex itself [21] is notably not substantially depleted, and only the p166 component (Tb927.11.3290) achieved an FDR < 0.01 following kDNA loss ($\log_2$ fold-change = -0.52 + /-0.12); indeed, kDNA is not required for assembly of this complex [32]. The mitochondrial carrier proteins [33], translocase of the inner mitochondrial membrane (TIM) [34], archaic translocase of the outer mitochondrial membrane (ATOM), and mitochondrial contact site and cristae organization system [35], were all increased in abundance (Fig 6D), perhaps compensating for defects in mitochondrial import and supporting the maintenance of mitochondrial membrane potential [13,36]. Proteins present in the outer mitochondrial membrane proteome (POMP), many of which remain otherwise uncharacterised [37], were also increased in abundance. Finally, the ER-membrane complex (EMC), previously connected to kDNA dependency [6], and now known to localise to the mitochondrial – ER interface [38], was increased in abundance (Fig 6D). Taken together, our results reveal ATP synthase complex remodelling associated with bi-allelic $\gamma ATPase$ mutation and resistance to both ATPase and kDNA-targeting drugs; oligomycin and acriflavine, respectively. These cells readily tolerate kDNA loss, which is associated with substantial mitochondrial proteome remodelling.

## Discussion

$\gamma$ATPase mutations in *T. brucei* are associated with kDNA loss and multidrug resistance and are also correlated with tsetse-fly independent mechanical transmission and geographical spread of these parasites beyond Africa. Here, we explore $\gamma$ATPase mutations and connections to kDNA loss in *T. b. brucei*. We precision-edited the native $\gamma ATPase$ gene, confirming that $L^{262}P$ and $A^{273}P$ mutants are resistant to the ATP synthase targeting drug oligomycin and to the kDNA-targeting drug, acriflavine. We also identified novel oligomycin-resistant aromatic amino-acid mutants replacing $M^{282}$. Quantitative proteomics analysis of homozygous $M^{282}$ mutants revealed specific depletion of ATP synthase $F_O$ components prior to kDNA loss. Following acriflavine-induced kDNA loss, confirmed to be complete by genome sequencing, we observed substantial mitochondrial proteome remodelling; the abundance of mitochondrial DNA and mRNA binding proteins was reduced while the abundance of proteins involved in mitochondrial import was increased. While we observed further depletion of ATP synthase $F_O$ components following kDNA loss, *c* subunit abundance was increased, perhaps reflecting accumulation of $F_1$-ATP synthase associated with only the *c*-ring of the $F_O$ moiety (see Fig 1D).

In terms of the origins of kDNA loss, we found that precision-editing could be used to generate both heterozygous and homozygous $\gamma$ATPase mutants in otherwise wild-type trypanosomes. While a heterozygous $M^{282}F$ edit failed to confer resistance to acriflavine, a homozygous $M^{282}F$ mutant was acriflavine-resistant and readily tolerated kDNA loss. We used quantitative proteomics to explore the impact of the homozygous $M^{282}F$ edit prior to kDNA loss and observed highly specific depletion of ATP synthase-associated proteins; possibly due to increased turnover when unassembled. Importantly,

we selected for edited cells using oligomycin rather than a DNA-damaging agent, avoiding direct selective pressure on the kDNA and likely reducing the potential for off-target mutations at this stage of the process. It is also worth noting in this regard that the widespread use of veterinary anti-trypanosomal drugs that target the kDNA, such as the DNA-damaging agents, ethidium bromide and isometamidium, could induce γATPase mutations and/or other mutations, and promote kDNA loss in the field [39].

A homozygous M$^{282}$F γATPase edit generated here in *T. b. brucei* was sufficient to confer kDNA dispensability, while a heterozygous edit was insufficient. This recessive effect with respect to kDNA dispensability suggested a dosage effect and prompted further consideration of naturally occurring non-synonymous γATPase mutations implicated in conferring kDNA dispensability; a heterozygous M$^{282}$L mutation in some isolates of *T. b. evansi* [14], a homozygous A$^{273}$P mutation in *T. b. equiperdum* [14], and a homozygous L$^{262}$P mutation in *T. b. brucei* [12]. Among these non-synonymous edits, we only failed to recover M$^{282}$L using oligomycin selection. Indeed, ectopic mutant γATPase expression assays in *T. b. brucei* also indicated that the M$^{282}$L mutation is insufficient to confer kDNA dispensability [12]. On the other hand, ectopic expression assays suggested that a single A$^{273}$P or L$^{262}$P allele may be sufficient to confer kDNA dispensability, and our demonstration that heterozygous A$^{273}$P or L$^{262}$P edits are sufficient to confer acriflavine resistance is consistent with this view. Homozygous editing described here is the first example of dual-allele oligo targeting, suggesting that this approach may be exploited to generate and assess further homozygous *γATPase* mutants.

We also used quantitative proteomics to elucidate the consequences of kDNA loss in homozygous M$^{282}$F γATPase mutants and observed extensive proteome remodelling in this case. Perhaps unsurprisingly, kDNA-binding proteins and mitochondrial RNA-processing factors were significantly depleted in the absence of mitochondrial DNA and RNA; residual editing complexes in kDNA-negative cells have been reported to retain function, however [40]. Mitochondrial membrane-associated transporters on the other hand were significantly increased in abundance, suggesting a boost in mitochondrial protein import capacity and inter-organellar trafficking capacity. Indeed, we found increased abundance of components of the ER membrane complex particularly intriguing, since we previously linked expression of this complex to kDNA dispensability in the absence of γATPase mutation [6]; this complex is now known to localise to the mitochondrial – ER interface [38]. These adaptations may compensate for mitochondrial import defects associated with changes in F$_1$F$_O$-ATP synthase assembly and maintenance of mitochondrial membrane potential. Similarly, these adaptations may reflect a response to depletion of other multiprotein complexes that contain kDNA-encoded proteins, such as respiratory complex I or the mitoribosome [13,36]. Proteome remodelling was not sufficient to recover a wild-type growth rate here, however, consistent with further adaptation thought to have occurred in *T. b. evansi* and *T. b. equiperdum* [9].

Notably, association of mutant F$_1$ with the inner mitochondrial membrane, perhaps proximal to the ATP/ADP carrier, is thought to be required to sustain mitochondrial membrane potential following kDNA loss [15,41]. Subunit *c* of the ATP synthase was selectively increased in abundance following kDNA loss, and although it was suggested that subunit *c* interacts with the calcium uniporter [25], the uniporter was reduced in abundance. We note here that analysis by native gel electrophoresis detected putative 'F1-*c*' complexes as a major ATP synthase assembly state in dyskinetoplastic *T. brucei* [41]. In the ATP synthase assembly pathway elucidated for other eukaryotes, subunit *a* (A6 in trypanosomes) is attached to F$_1$-*c* before assembly with the F$_O$ part [42]. We therefore suggest that, in the absence of the kDNA-encoded subunit A6, ATP synthase assembly beyond F$_1$-*c* is impaired; a hypothesis that could be tested in the future. Other adaptations may reflect disrupted communication between the kDNA and nuclear DNA. For example, PUF9 target 1, a kDNA replication-associated peptidase [28], was increased in abundance while the RNA-binding protein PUF9, involved in nuclear cell cycle regulation [26], was reduced in abundance. The telomere-telomerase complex was also increased in abundance. Taken together, these adaptations reveal a remarkable connectivity between the ATP synthase and other mitochondrial, and even other cellular, complexes and compartments.

In conclusion, *T. brucei* cells with a bi-allelic *γATPase* defect assemble a remodelled ATP synthase complex, and tolerate kDNA-loss, accompanied by substantial mitochondrial proteome remodelling. Proline mutations with the potential

to disrupt helical structure at $L^{262}$ or $A^{273}$ in the γATPase [12,14], or bulky aromatic residue mutations at $M^{282}$, introduce defects analogous to a broken camshaft at the core of this ATP synthase rotary motor. Our findings yield new insights into the origins and consequences of kDNA loss, with implications for the evolution of trypanosome sub-species that have global veterinary impacts.

## Methods

### *T. brucei* growth and *γATPase* gene editing

Bloodstream form *T. b. brucei* Lister 427 cells were grown in HMI-11 (Gibco) supplemented with 10% fetal bovine serum (Sigma) at 37°C and with 5% $CO_2$ in a humidified incubator. For site saturation mutagenesis using oligo-targeting, a degenerate ssODN was transfected in duplicate by electroporation with a Nucleofector (Lonza), and a human T-cell kit (Lonza), with the Nucleofector set to Z-001 (Amaxa). Briefly, we used 40 µg of the ssODNs in 10 µl of 10 mM Tris-Cl, pH 8.5, mixed with 25 million cells in 100 µl transfection buffer. 200 nM oligomycin was applied 6 h after transfection. DNA was isolated 7 d later. The *γATPase* fragment was amplified by PCR using Q5 high fidelity DNA polymerase (New England Biolabs) as per the manufacturer's instructions, and primers 1 and 2. Annealing was at 63°C and elongation was for 30 s. PCR products were purified using a Qiagen PCR purification kit. For specific mutagenesis using oligo targeting, a specific ssODN was transfected in duplicate with 10 million cells, and 200 nM oligomycin was applied 6 h after transfection. Oligomycin-resistant cultures were sub-cloned by serial dilution in 96-well plates 5 d later and DNA was extracted from the clones. The *γATPase* fragment was amplified by PCR as above but using primers 1 and 3 in this case. The products were Sanger sequenced using primer 4 at Azenta Life Sciences. To induce kDNA loss, cells were grown in the presence of 1.25 nM acriflavine for 7 days and then subcloned.

### Dose-response assays

To determine the Effective Concentration of drug to inhibit growth by 50% ($EC_{50}$), cells were plated in 96-well plates at 1 x $10^3$ cells/ml in a 2-fold serial dilution of selective drug. Plates were incubated at 37°C for 72 h. 20 µl resazurin sodium salt (AlamarBlue, Sigma) at 0.49 mM in PBS was then added to each well, and plates were incubated for a further 6 h. Fluorescence was determined using an Infinite 200 pro plate reader (Tecan) at an excitation wavelength of 540 nm and an emission wavelength of 590 nm. $EC_{50}$ values were derived using Prism (GraphPad).

### Mass spectrometry

Approx. 5 x $10^7$ PBS-washed cells were suspended in 100 µL of TBA (5% SDS, 100 mM triethylammonium bicarbonate); triplicate samples for each experiment and control cells. Total cell extracts were submitted to the Fingerprints Proteomics Facility at the University of Dundee and processed by trypsin: µBCA (bicinchoninic acid), strap processed, quality controlled, and peptide quantified by Micro-BCA assay (Thermo Scientific). 3 µl of each sample was processed using S-Trap Micro columns (Protifi) where proteins were reduced, alkylated and digested overnight at 37°C at 1:40 enzyme-to-substrate. A second digest was repeated for 6 h the following day. For mass spectrometry analysis, digested peptides (200 ng) were run on an Astral Orbitrap Mass Spectrometer (Thermo Scientific) coupled to a Vanquish Neo UHPLC system (Thermo Scientific). Buffer conditions used Buffer A (0.1% formic acid) and Buffer B (80% acetonitrile in 0.1% formic acid). Flow was 60 µl/min and loading volume was set at automatic. Peptides were initially trapped on a PepMap Neo C18 column (5 µm, 300 µm x 5 cm) and then separated on an Easy-Spray PepMap RSLC C18 column (2 µm, 150 µm x 15 cm) (Thermo Scientific). Columns was kept at a constant temperature of 50°C and a source voltage of 2.0 kV. Full MS scan was performed in data-independent acquisition (DIA) mode with an m/z range of 380–980 with orbitrap resolution 2400000, Automatic Gain Control (AGC) target of 500% and a maximum injection (IT) of 3 ms. MS scans were followed by MS/MS DIA using the following parameters; scans of isolation window of 2.0 m/z unit and window overlap set at 0

m/z. Normalised collision energy was set to 25%. Data for MS scans were acquired in profile mode with MS/MS DIA scan events being acquired in centroid mode.

### Proteomic data analysis

We generated a spectral library based on the predicted protein sequences for *T. brucei* TREU927 sourced from TriTrypDB (version 51) [43]. The raw mass spectrometry data were processed using DIA-NN (version 2.2.1) [44] and analysed using default settings. Initial QC analysis, median normalisation and missing value imputation was performed with the project utility python package (https://github.com/mtinti/ProjectUtility). Differential expression analysis was performed using the limma package in R [45]. A linear model was fitted using the lmFit() function, followed by Empirical Bayes Statistics for Differential Expression computed with the ebayes() function. Adjusted p-values were calculated using the topTable function with the Benjamini & Hochberg (BH) correction method.

### High-throughput sequencing

For analysis following site saturation mutagenesis, PCR amplicons were sequenced at the Beijing Genome Institute (BGI) on a DNBseq platform with 150 base paired-end reads as described previously [16] and codon-based read counts were derived using the OligoSeeker software [46]. Whole genome sequencing data were analysed with alignment to the *T. b. brucei* reference genome v46 clone 427_2018 supplemented with 427 maxi and minicircle sequences [2,43]. The alignment and read counts were performed with the automated snakemake [47] pipeline myRna-seq [48]. Read coverage was extracted from the BAM files using a bin size of 150 with the bamCoverage function from deepTools (v 3.5.6). The circular visualisation was performed with the pyCirclize (1.6) Python package (https://github.com/moshi4/pyCirclize).

### Microscopy

To identify clones lacking kDNA, cells were fixed in 1% paraformaldehyde (PFA) for 15 min, washed twice in PBS and resuspended in water with 1% bovine serum albumin (BSA). Cells were attached to a 12-well 5 mm slide (Thermo Scientific) by drying overnight. After rehydration in PBS for 5 min, slides were mounted in Vectashield with DAPI and sealed under a coverslip. Cells were viewed at 63x magnification with oil immersion on a Zeiss Axiovert 200 M microscope with Zen Pro software (Zeiss). For mitochondria morphology visualisation, cells were stained with 100 nM Mitotracker red CMXRos (Invitrogen) for 5 min at 37°C prior to fixation in 3% PFA for 15 min, washed in PBS then resuspended in water with 1% BSA. Cells were attached to poly-lysine-coated coverslips for 4 h at room temperature then stained with DAPI for 30 min before mounting in Vectashield (without DAPI) and sealing to a glass slide. For wide-field microscopy, cells were imaged as z-stacks (0.2 μm) at 100x magnification with oil immersion on the same microscope described above. For super resolution microscopy, cells were imaged as z-stacks (0.1–0.2 μm) at 63x magnification on a Leica Stellaris 8 inverted confocal microscope equipped with Power HyD detectors and subjected to adaptive deconvolution using the integrated Leica LIGHTNING algorithm for super-resolution microscopy. Images were analysed using Fiji v1.5.2e.

### MitoTracker staining and flow cytometry

Live *T. brucei* cells ($1x10^6$/ml) were incubated with 100 nM MitoTracker Red CMXROS (Molecular Probes) at 37°C for 5 min. Cells were fixed with 1% of paraformaldehyde (PFA) at 37°C for 15 min, then washed and resuspended in cold PBS and stored at 4 °C. MitoTracker fluorescence intensity was measured using a CytoFlex S flow cytometer (Beckman Coulter) and analysed using FlowJo v10.10. Forward scatter area (FSC-A) versus forward scatter height (FSC-H) was used to exclude cell aggregates.

## Supporting information

**S1 Fig. Dose-response curves. (A)** For oligomycin, measured in duplicate. **(B)** For acriflavine, measured in duplicate.
(TIF)

**S2 Fig. Proteomics analysis of homozygous γATPase M$^{282}$F mutants with and without kDNA.** Subunits of the $F_1$ and $F_o$ γATPase components are highlighted. Averages from three replicates; n = 6768 proteins; clone 1.2 is shown in Fig 5A.
(TIF)

**S3 Fig. Gallery of super resolution microscopy images showing wild-type *T. b. brucei* and homozygous γATPase M$^{282}$F mutant cells lacking kDNA.** Other details as in Fig 5B.
(TIF)

**S1 Data. Oligonucleotides and proteomics data.** Sheet 1: Oligonucleotides and primers used in this study. Sheet 2: Proteomics data – M$^{282}$F kDNA positive parent v wild-type. Sheet 3: Proteomics data – M$^{282}$F kDNA negative clone 1.1 v kDNA positive parent. Sheet 4: Proteomics data – M$^{282}$F kDNA negative clone 1.2 v kDNA positive parent. Sheet 5: Proteomics data – M$^{282}$F kDNA negative clone 2.1 v kDNA positive parent. Sheet 6: Proteomics data – M$^{282}$F kDNA negative clone 2.2 v kDNA positive parent.
(XLS)

## Author contributions

**Conceptualization:** Melanie Ridgway, David Horn.

**Data curation:** Michele Tinti.

**Formal analysis:** Melanie Ridgway, Douglas O. Escrivani, Achim Schnaufer.

**Funding acquisition:** David Horn.

**Investigation:** Melanie Ridgway, Douglas O. Escrivani, Markéta Novotná, Amy Wood.

**Project administration:** David Horn.

**Supervision:** Melanie Ridgway, David Horn.

**Visualization:** Michele Tinti.

**Writing – original draft:** Melanie Ridgway, Douglas O. Escrivani, Achim Schnaufer, David Horn.

**Writing – review & editing:** Melanie Ridgway, Douglas O. Escrivani, Markéta Novotná, Amy Wood, Michele Tinti, Achim Schnaufer, David Horn.

## Acknowledgments

We thank Gustavo Bravo Ruiz for assistance with visualising GO-term profiles.

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
