## [Decision Letter · Decision Letter 0]

29 Jan 2026

Origins and consequences of kinetoplast loss in trypanosomes

PLOS Pathogens

Dear Dr. Horn,

Thank you for submitting your manuscript to PLOS Pathogens. After careful consideration, we feel that it has merit but does not fully meet PLOS Pathogens's publication criteria as it currently stands. Therefore, we invite you to submit a revised version of the manuscript that addresses the points raised during the review process.

We look forward to receiving your revised manuscript.

Kind regards,

Cynthia Y. He

Academic Editor

PLOS Pathogens

Dominique Soldati-Favre

Section Editor

PLOS Pathogens

Sumita Bhaduri-McIntosh

Editor-in-Chief

PLOS Pathogens

orcid.org/0000-0003-2946-9497

Michael Malim

PLOS Pathogens

orcid.org/0000-0002-7699-2064

**Additional Editor Comments:**

All three reviewers agreed that the work presented interesting new data that may contribute to the understanding of kDNA loss and mitochondrial biology in general. The main critique is that the mutants weren’t functionally or morphologically characterized, and the proteomics results lack experimental confirmation. Additional studies to characterize mitochondrial morphology and functions in the mutants are suggested. Corroborating changes in targeted pathways and the formation of protein complexes experimentally would enhance the manuscript.

**Journal Requirements:**

At this stage, the following Authors/Authors require contributions: Melanie Ridgway, Douglas Escrivani, Marketa Novotna, Amy Wood, Michele Tinti, Achim Schnaufer, and David Horn. Please ensure that the full contributions of each author are acknowledged in the "Add/Edit/Remove Authors" section of our submission form.

Potential Copyright Issues:

i) Figure 1A. Please confirm whether you drew the images / clip-art within the figure panels by hand. If you did not draw the images, please provide (a) a link to the source of the images or icons and their license / terms of use; or (b) written permission from the copyright holder to publish the images or icons under our CC BY 4.0 license. Alternatively, you may replace the images with open source alternatives. See these open source resources you may use to replace images / clip-art:

**Reviewers' Comments:**

Reviewer's Responses to Questions

**Part I - Summary**

Reviewer #1: This manuscript investigates how mutations in the gamma subunit of the mitochondrial F₁F₀-ATP synthase (gATPase) facilitate survival of Trypanosoma brucei in the absence of the kinetoplast DNA (kDNA). The study builds on the observation that certain gATPase mutations render T. brucei resistant to drugs targeting mitochondrial functions and enable life without kDNA. By employing targeted gene editing and selection with the ATP synthase inhibitor oligomycin, the authors isolate and characterize novel and known gATPase mutations (M282F, M282W, M282Y, L262P, A273P) and analyze their biochemical, proteomic, and cellular consequences.

This work provides an important contribution to our understanding of mitochondrial genome independence in eukaryotes and explores the functional consequences of mitochondrial genome loss, which has implications for parasite biology, transmission, and drug resistance. However, some aspects of the mechanistic interpretation and data presentation require further clarification and experimental support.

Major Claims by the Authors

1. Mutations in the gamma subunit enable viability without kDNA including the newly found M282F homozygous mutant

2. Study aims to explore the origins and consequences of kDNA loss

3. Use of oligomycin for enrichment of gamma ATPase mutants following oligo-directed mutagenesis

4. Homozygous M282F mutants resist acriflavine and survive kDNA loss

5. Proteomic analyses show mitochondrial remodeling associated with kDNA loss

o Pre-kDNA loss: Specific depletion of ATP synthase-associated proteins (excluding F₁ subunits).

o Post-kDNA loss: Loss of kDNA-binding proteins and mitochondrial RNA processing factors; increase in mitochondrial membrane-associated transporters.

6. Conclusion

o Cells with homozygous M282F mutations assemble a remodeled ATP synthase, tolerate kDNA loss, and undergo substantial mitochondrial proteome remodeling.

Reviewer #2: While the flexible mitochondrial biology of Trypanosoma brucei has been studied in detail, knowledge about how mutations that render the cells susceptible to kDNA loss and their full impact during drug treatments are understudied. In this body of work, the authors use a rapid and specific precision editing tool based upon oligo-targeting first reported in the field by the Horn lab. They combined this with a selection using 200 nm oligomycin. The authors generated mutants against known gamma ATPase mutations associated with kDNA dispensability and recover new mutations at the position M282. The methodology is robust with good internal controls to distinguish heterozygote and homozygous mutants.

The authors are laying the foundational tools for futures studies on analyzing mutations that occur in the wild and could drive drug resistance. There is a lot of interesting data that was generated but perhaps not fully analyzed. The heterogyzous mutant seems important for the trajectory of the population becoming kDNA dispenable but was not fully analyzed.

Reviewer #3: This study demonstrates that specific mutations in the nuclear-encoded γ subunit of the mitochondrial F₁F₀-ATP synthase enable Trypanosoma brucei to tolerate complete loss of kinetoplast DNA. Using precision oligo-targeting, the authors identify a novel homozygous γATPase M282F mutation that confers reduced sensitivity to both oligomycin and the kDNA-targeting drug acriflavine, and show that this mutation permits acriflavine-induced elimination of the mitochondrial genome. Quantitative proteomics reveals that γATPase mutation leads to selective remodelling of the ATP synthase complex prior to kDNA loss, followed by extensive mitochondrial proteome reprogramming after kDNA loss, including depletion of kDNA-binding and RNA-processing factors and upregulation of mitochondrial membrane transport systems.

While the study does not provide fundamentally new mechanistic insight into how γATPase mutations uncouple the ATP synthase, the proteomic analyses generate a valuable dataset that may be useful for further exploration of cellular adaptations to mitochondrial DNA loss.

**Part II – Major Issues: Key Experiments Required for Acceptance**

Please use this section to detail the key new experiments or modifications of existing experiments that should be absolutely required to validate study conclusions.required to validate study conclusions.

Reviewer #1: 1. Oligomycin Resistance and Wild-Type gamma ATPase Contribution

The manuscript should explore whether the level of oligomycin resistance correlates with residual wild-type gamma ATPase expression in heterozygous strains. Such analysis would provide insights into the dosage sensitivity and functional threshold of the mutant protein.

2. Mitochondrial Morphology and Organelle Interaction

How do these mutations affect:

• Mitochondrial ultrastructure?

• ER-mitochondria contact sites?

• Electrochemical potential across the mitochondrial membrane?

These aspects are central to mitochondrial function and may be altered in kDNA-lacking cells. The authors should consider EM or confocal imaging, and membrane potential assays.

3. ATP Synthase Assembly Pathway

The authors suggest that assembly is arrested post-F₁–c-ring attachment, in analogy to other eukaryotes. This hypothesis is testable using blue native PAGE. Inclusion of such data or acknowledgment of this as a future direction would strengthen mechanistic claims.

4. TAC Components and kDNA Loss

Given the complete loss of kDNA in homozygous mutants, the status of TAC (tripartite attachment complex) proteins should be addressed. Are these proteins depleted? Their inclusion in the proteomic analysis (especially in Figure 5) would provide a useful internal control and functional insight.

5. Mitochondrial Import Machinery

The increase in membrane-associated transporters is noted, but specific components and the extent of change should be detailed in the main text (not solely in figures). Which import components are increased, and by how much?

Figure-Specific Critiques

Figure 1D

• Structural labeling is incomplete. Clearly label F₁ and F₀ domains.

• Mutation positions (e.g., M282F) should be mapped explicitly also in the stick figure

• Color coding lacks explanation; provide a legend or rationale.

Figure 3

• Clarify the meaning of n in the legend; does not refer to replicates?

• Justify inclusion of both log₂ fold-change and –log₁₀ FDR. Are both dimensions necessary?

• Consider whether this figure is redundant with Figure 5. If retained, explain the complementary value.

Figure 4A

• Discrepancy noted between DNA channel and merged image.

• A single-cell image is insufficient to claim kDNA loss. Include a field with multiple cells for representativeness.

Figure 5

• The presentation is inconsistent and cluttered. Multiple visualization types could/should be unified.

• Cohort terms (Figure 5E) require definition. 400+ proteins for "kinetoplast" are a lot? Is that real? Are functional overlaps considered, i.e. the ribosome etc.?

• Indicate what red/blue colors represent.

• Include TAC proteins to assess stability of mitochondrial structural elements.

• Revise labeling of "increased" and "decreased" with fold enrichment to improve clarity.

• PUF9 Target 1 Speculation: The discussion suggests that kDNA loss disrupts kDNA–nuclear genome communication. This needs clarification. What is the functional relevance of PUF9 targets in this context? How does their regulation support the proposed communication axis?

• Scope and Novelty: While the experiments are technically sound and well-conceived, the manuscript would benefit from a clearer articulation of what new insights this study adds over prior work. Specifically:

o What do the newly identified mutations (e.g., M282F) reveal beyond earlier L262P and A273P studies?

o How does this work advance our understanding of the origins (not just consequences) of kDNA loss?

• Comparative Context: The authors may consider placing their findings in the broader context of mitochondrial genome loss in other systems (e.g., Saccharomyces cerevisiae “petite” mutants) to highlight evolutionary or mechanistic parallels.

Reviewer #2: 1. In the current study it appears that the kDNA loss does not occur until after drug pressure from acriflavine. This is not a compound that would be used in practice though. If one is considering what is driving kDNA loss and drug resistance in the field perhaps using a drug more relevant to human treatments is warranted. With proof of principle in place with acriflavine, the authors should use a drug(s) in use and evaluate the outcome of their homozygous M282F mutant (perhaps even the hetrozygote).

2. Authors should provide more details on the proteomics experiment of homozygous M282F mutant vs WT. There is clearly a subset of proteins other than F0 or F1 subunits that are also impacted when generating this mutant cell line. Did the authors perform proteomics on the M282F hetrozygote to identify significant changes form WT that were emerging to drive the population toward becoming kDNA dispensible? This progression would be then interesting to correlate with the different responses to acriflavine and oligomycin dose-response data.

3. Similarly because selection of the ATPase mutants is already an experiment – the authors should report percent loss of kDNA compared to WT for each of the precision oligo edited clones that were generated (ie. prior to sub-lethal treatment with acriflavine or other drug).

4. Could the authors please rewrite the abstract. The authors’ summary was more informative/clear than the abstract (just lacked some specifics). For example in the abstract pre-kDNA-loss is introduced out of context; proposed a remodelled ATP suynthase complex – was this isolated and characterized? Rewriting would provide better clarity.

5. It would be beneficial if the authors could focus the discussion more on how homozygous vs heterozygous mutations could drive events in the wild. This is interesting and directly related to the problem of emerging subspecies and drug resistance….and related this to current drug classes that are used or overused in practice.

Reviewer #3: Although the genetic and proteomic analyses strongly suggest altered ATP synthase function and mitochondrial physiology in the gamma ATPase mutants, the study remains largely descriptive. The manuscript would be strengthened by direct functional measurements, such as mitochondrial membrane potential, ATP levels, or ATPase activity, before and after kDNA loss. Such data would more firmly link the observed molecular changes to mitochondrial function.

The quantitative proteomics provide a rich dataset documenting extensive mitochondrial remodelling following kDNA loss. However, the functional significance of many of the observed changes remains unclear (for example, increased abundance of TOM, TIM, MICOS complexes, and various mitochondrial carrier proteins). Experimental validation of selected pathways or protein complexes highlighted in the proteomic analysis would strengthen conclusions regarding their adaptive or mechanistic roles.

The growth defect observed in kDNA-negative mutants is noted but not analysed further. Additional characterization of this phenotype could provide important insight into the biological costs associated with kDNA loss and help contextualize the adaptive significance of these mutations.

**Part III – Minor Issues: Editorial and Data Presentation Modifications**

Reviewer #1: (No Response)

Reviewer #2: 1. Please define kinetoplast more clearly. In some cases it sounds as if this is a separate organelle, in other instances it is stated “loss of the kinetopalst” – so loss of an organelle or loss of the DNA? Throughout the manuscript there are instances where kineoplast loss is used and other where kDNA loss is preferred.

2. eliminate use of “approx.”

3. Discussion of multidrug resistant – please be more precise by indicating which two classes of drugs are involved with the resistance, or are these parasites resistant to all drugs in use for treatments?

4. abbreviation for kinetoplast DNA is defined in Introduction – it is only needed once but is reintroduced several times.

5. Please state the doubling times of the different clones that were generated.

6. Authors should acknowledge TritrypDB for information they access – you reference the database in figure 5 legend – full acknowledgment is appropriate.

7. Figure 1D – the authors indicate that there are predicted conformational changes due to the precision editing mutants. Two are obvious a third is less obvious. Could the authors provide additional text to describe the difference since it is hard to see this in a status 2D image.

8. Could the authors link the conformational changes to M282F proteomic outcomes – are there similarities/differences that arise from the L262P and A273P structural changes when compared the current dataset for M282F? Reviewer recognizes that this data might not have been collected in the previous studies and does not think this type of additional proteomics studies of the other mutants are warranted in the current study.

9. awkward wording that described which strain was used for sequencing – just state the strain.

10. End of paragraph related to Fig3 – statements was…. Thus, proteomic analysis revealed highly specific depletion of subunits of the FO component of the T. b. brucei ATP synthase in homozygous M282FTTT mutants pre kDNA loss. Perhaps it would be better to state it as prior to acriflavine treatment?

11. Could the authors report how long it took to achieve stable populations following the sublethal treatment with acriflavine and the dilution cloning process – they mentioned slower doubling times….were the cells considered fully recovered after dilution cloning?

Comments on Figure and Legends:

1. Fig1A: - Nice diagram outlining approach, however in the flask designated oligomycin-resistant cell, why are all the cells blue. As diagramed the approach should yield red and green resistant cells from the precision editing.

2. Fig1B; what does the red outline square represent?

3. Fig1D: what does the vertical line represent in the alphafold portion of the figure?

4. Fig4A: authors should outline the cells or provide phase images. Current images look as if the nucleus is outside of the cell body.

5. Fig4B: the labelling of the Mini sequence data is a little misleading – is their nomenclature that can be added to indicate those are indeed different classes of sequences?

6. Fig4B legend: please specify exactly which clone was used for the sequence analysis.

7. Fig. 4C: it is very difficult to see data because of the grey scale chosen. Please improve by using a colored scale. It looks like there might be some maxicircle sequence retained in one of the clones???

8. Fig5E: The boxes and whiskers are very difficult to see in the current figure.

9. Figure5E legend: indicate what the two colors represent.

Reviewer #3: The title appears more suitable for a review article than for a primary research paper. In particular, the term “origins” may imply a broad evolutionary origin of organisms lacking mitochondrial DNA, whereas this study focuses on the identification of three γATPase mutations and experimental validation of one of them. Similarly, the term “consequences” may be somewhat overstated. The consequences described here are largely inferred from proteomic datasets. While valuable, these data provide limited direct insight into how cells functionally adapt to the loss of mitochondrial DNA.

The authors state that the homozygous M282F mutant is resistant to acriflavine, yet kDNA loss is induced using this same drug. Given that similar EC₅₀ values are observed for heterozygous L262P and A273P mutants, it appears that this is how “resistance” is defined in this context. However, acriflavine is then used to induce kDNA loss in an acriflavine-resistant strain, which is difficult to reconcile. Clarification of the definition of “resistance” would therefore be helpful. In addition, it would be informative to indicate the acriflavine sensitivity of naturally occurring T. evansi and T. equiperdum, which are expected to be kDNA-independent.

Can the whole-genome sequencing data be used to assess whether compensatory mutations arise in the nuclear genome following kDNA loss? A brief comment on this possibility would strengthen the manuscript.

The statement that doubling time increased by 47% is somewhat abstract. Reporting the actual doubling times and stating that growth is slowed approximately 1.5-fold would be clearer and more intuitive.

The strain labels used in the supplementary tables (e.g. S22, S24, S312, S315) do not clearly correspond to the strain descriptions in the main text or figures, which use +kDNA and −kDNA designations. Clarification of this nomenclature is needed.

Which specific mitochondrial carrier proteins are upregulated? Can any functional adaptations be inferred from these changes?

Several sections refer to decreased mitochondrial membrane potential in −kDNA cells. Can this be experimentally measured to directly support this conclusion?

The depletion of ATP synthase Fo components prior to kDNA loss is a key observation. Is this more likely due to protein instability (e.g. degradation of unassembled subunits) or to regulated changes in gene expression? Clarification or discussion of this point would be helpful.

The authors suggest that F₁ remains attached to the c-ring in the absence of other Fo components. Could this be tested experimentally, for example using blue native PAGE to assess ATP synthase subcomplexes?

PLOS authors have the option to publish the peer review history of their article (what does this mean? ). If published, this will include your full peer review and any attached files.). If published, this will include your full peer review and any attached files.

**Do you want your identity to be public for this peer review?** For information about this choice, including consent withdrawal, please see our For information about this choice, including consent withdrawal, please see our Privacy Policy ..

Reviewer #1: No

Reviewer #2: No

Reviewer #3: No

**Figure resubmission:**

**Reproducibility:**



---

## [Decision Letter · Decision Letter 1]

14 Mar 2026

Dear Dr. Horn,

We are pleased to inform you that your manuscript 'Genetic origins and proteomic consequences of kinetoplast loss in trypanosomes' has been provisionally accepted for publication in PLOS Pathogens.

Before your manuscript can be formally accepted you will need to complete some formatting changes, which you will receive in a follow up email. A member of our team will be in touch with a set of requests. We would also like you to consider the minor changes suggested by Reviewer 2.

Best regards,

Cynthia Y. He

Academic Editor

PLOS Pathogens

Dominique Soldati-Favre

Section Editor

PLOS Pathogens

Sumita Bhaduri-McIntosh

Editor-in-Chief

PLOS Pathogens

orcid.org/0000-0003-2946-9497

Michael Malim

Editor-in-Chief

PLOS Pathogens

orcid.org/0000-0002-7699-2064

Reviewer Comments (if any, and for reference):

Reviewer's Responses to Questions

**Part I - Summary**

Reviewer #1: (No Response)

Reviewer #2: For this resubmission, the authors provide evidence that specific mutations in the nuclear-encoded γ subunit of the mitochondrial F₁F₀-ATP synthase enable Trypanosoma brucei to tolerate complete loss of kinetoplast DNA. Specifically, they use precision oligo-targeting, to generate a novel heterozygous and homogyzous γATPase M282F mutation. Only the homozygous mutant confers reduced sensitivity to the kDNA targeting drug acriflavine and demonstrate the homozygous mutation permits acriflavine-induced elimination of the kDNA. The authors use high resolution quantitative proteomics to further demonstrate selective remodelling of the ATP synthase complex prior to kDNA loss, followed by extensive mitochondrial proteome reprogramming after kDNA loss.

Reviewer #3: I appreciate the authors’ efforts to address the reviewers’ comments and have no further remarks.

**Part II – Major Issues: Key Experiments Required for Acceptance**

Please use this section to detail the key new experiments or modifications of existing experiments that should be absolutely required to validate study conclusions.required to validate study conclusions.

Reviewer #1: (No Response)

Reviewer #2: No Major issues identified.

Reviewer #3: (No Response)

**Part III – Minor Issues: Editorial and Data Presentation Modifications**

Reviewer #1: (No Response)

Reviewer #2: The authors address essentially all of the reviewers’ comments by adding additional functional data to assess mitochondrial membrane potential under the various WT and mutant conditions and modified images. Additionally, the authors added clarifying language to the text especially related to the impact on known protein complexes. They have made a clear statement on the preservation of the TAC complex (except for p166), clarified language surrounding a hypothesized ATP synthase assembly pathway and highlight additional ATPas associated proteins in their proteomic analyses.

Lastly, two reviewers thought the heterozygous mutant was significant. The authors briefly addressed the heterozygosity issue with the following statement in the manuscript ”Since differential expression of mutant alleles could impact the behaviour of heterozygous mutants, we favoured the analysis of homozygous mutants.”

It would be beneficial if the authors could add additional content especially related to their more extended explanation to reviewers: “Even if we were able to determine the relative levels of wild-type and mutant gATPase protein in the heterozygous strains, we would be unable to determine relative levels in individual cells, which could impact oligomycin resistance and other phenotypes.”

Can the authors indicate what is known in trypanosomes about expression from different alleles (in this case a WT vs mutant allele) or is this an area that is completely unexplored in their field? An expanded discussion on this would be ideal.

Reviewer #3: (No Response)

PLOS authors have the option to publish the peer review history of their article (what does this mean? ). If published, this will include your full peer review and any attached files.). If published, this will include your full peer review and any attached files.

**Do you want your identity to be public for this peer review?** For information about this choice, including consent withdrawal, please see our For information about this choice, including consent withdrawal, please see our Privacy Policy ..

Reviewer #1: No

Reviewer #2: No

Reviewer #3: No

---

## [Editor Report · Acceptance letter]

Dear Dr. Horn,

We are delighted to inform you that your manuscript, "Genetic origins and proteomic consequences of kinetoplast loss in trypanosomes," has been formally accepted for publication in PLOS Pathogens.

Best regards,

Sumita Bhaduri-McIntosh

Editor-in-Chief

PLOS Pathogens

orcid.org/0000-0003-2946-9497

Michael Malim

Editor-in-Chief

PLOS Pathogens

orcid.org/0000-0002-7699-2064